# A comparative analysis of co-contraction indices using synthetic EMG data: Implications for selection and interpretation

Hannah D. Carey[1], Friedl De Groote[2], Andrew Sawers[1]*

**1** Department of Kinesiology and Nutrition, University of Illinois Chicago, Chicago, Illinois, United States of America, **2** Department of Movement Sciences, KU Leuven, Leuven, Vlaams-Brabant, Belgium

* asawers@uic.edu

## Abstract

Co-contraction—the simultaneous activation of opposing muscles—influences movement efficiency, joint stability, and motor learning. While consensus exists for EMG acquisition and processing, comparable guidance for quantifying co-contraction is lacking. This study evaluated the behavior and interrelationships of six commonly used co-contraction indices (CCIs) to develop practical recommendations for their selection, calculation, and interpretation. Synthetic EMG-like signals were generated and used to evaluate CCI behavior across a range of conditions that would be difficult to achieve experimentally. Based on their formulas and observed behavior, CCIs were sorted into three categories: *shape-based, amplitude-driven, and temporal indices.* Shape-based CCIs increase when two EMG signals have similar shapes, regardless of amplitudes. Amplitude-driven CCIs increase when activation is high in both muscles. Temporal CCIs increase as the duration of EMG overlap increases, regardless of signal shape or amplitude. Correlation analyses showed stronger associations within-category than between-category, supporting the proposed classification scheme. CCI behavior yielded three principal findings, each paired with a practical recommendation. First, EMG amplitude normalization techniques altered co-contraction estimates, and the effect varied by index. Researchers should therefore test whether their conclusions hold across normalization methods. Second, because CCIs differ in scale and theoretical maxima, their values are not directly comparable across indices. Comparisons should instead focus on relative trends interpreted within each index's bounds. Third, each CCI category was sensitive to different EMG features (e.g., amplitude versus shape). The choice of CCI should therefore align with hypothesized differences in EMG signals—use shape-based CCIs when waveform similarity is of interest, and amplitude-driven CCIs when differences in activation magnitude are expected. These results provide initial guidance for selecting, calculating, and interpretating CCIs, and they establish a framework

**Data availability statement:** All relevant data for this study are publicly available from the Zenodo repository (https://doi.org/10.5281/zenodo.20058872).

**Funding:** The authors disclosed receipt of the following financial support for the research, authorship, and/or publication of this article: Research reported in this publication was supported by the Department of Defense under Award No. HT9425-24-1-0212. The content is solely the responsibility of the authors and does not necessarily represent the official views of the Department of Defense. The funders had no role in study design, data collection and analysis, decision to publish, or preparation of the manuscript.

**Competing interests:** The authors have declared that no competing interests exist.

for testing the robustness of these theoretical findings using experimental EMG from diverse tasks, muscle pairs, and populations.

---

## 1. Introduction

Muscle co-contraction—the simultaneous activation of agonist and antagonist muscles crossing the same joint and acting in the same plane [1,2]—is central to understanding and controlling human movement. Co-contraction has been associated with movement inefficiency [3,4], early motor learning [5–8], robust resistance to mechanical disturbances [9], altered reflex responses [10–13], increased joint stiffness and stability [14–18], and reduced stress on joint ligaments [19]. Although there is extensive consensus on EMG collection, processing, and analysis [2,20–24], best practices for quantifying and interpreting co-contraction have received comparatively less attention. Systematic and scoping reviews document substantial diversity in co-contraction indices (CCI) and suggest that heterogeneity in populations, tasks, and methods impedes cross-study synthesis and clear guidance on index selection [25–27]. Experimental comparisons of CCIs have yielded valuable insights [18,28–30], but most evaluate only two indices, limiting their scope. The generalizability of these experimental findings is further constrained by study-specific EMG recordings and the noise inherent to EMG data. Synthetic EMG helps address these limitations by enabling targeted simulations of specific signal features and more precise evaluation of CCI behavior than are feasible with experimental data. For example, synthetic (i.e., simulated) datasets have been used to test how signal-to-noise ratio and preprocessing steps influence CCI outcomes [31]. A comprehensive assessment of CCI behavior under well-defined, controlled conditions using synthetic data is needed to develop actionable recommendations for CCI selection and interpretation. The purpose of this study was to characterize the behavior and interrelationships of six commonly used CCIs and develop practical guidance for their selection, calculation, and interpretation.

## 2. A brief description, comparison, and categorization of common co-contraction indices

Before reviewing, analyzing, and comparing six common co-contraction indices (CCI) [1,32–34], we first define some basic nomenclature. $EMG_1(t)$ and $EMG_2(t)$ represent the amplitude normalized electromyography (EMG) signals recorded from an agonist—antagonist muscle pair at time $t$. At each time point $t$, the higher and lower values of $EMG_1(t)$ and $EMG_2(t)$ are denoted as $EMG_{high}(t)$ and $EMG_{low}(t)$, respectively.

### 2.1. The Simple ratio Co-Contraction Index ($CCI_{SR}$)

One of the earliest and simplest methods for quantifying co-contraction is the ratio of antagonist to agonist EMG activity (Eq. 1; Fig 1A) [35]. Simple ratio CCIs can be calculated using any number of EMG metrics derived from each muscle, including the peak, root mean squared, or integral of agonist and antagonist activity [36]. Because we often consider EMG signals with more complex shapes than those that can be

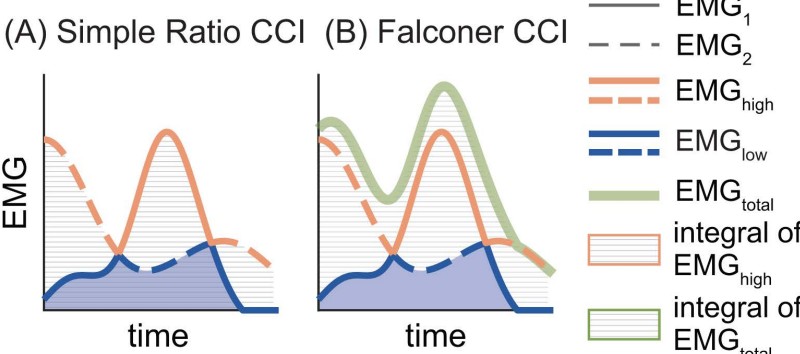

**Fig 1. Visualizing the Simple Ratio (A) and Falconer-Winter (B) co-contraction indices (CCIs).** Solid and dashed lines represent $EMG_1$ and $EMG_2$, respectively. At each time point, the blue and orange lines correspond to the low and high EMG signals, respectively, while the solid green line in panel B shows the total EMG (i.e., $EMG_1 + EMG_2$). The blue shaded area represents the integral of $EMG_{low}$, which serves as the numerator in both indices (see Eqn. 1–2). In panel **(A)**, the gray hatched region denotes the integral of $EMG_{high}$, used as the denominator in the Simple Ratio CCI (Eqn. 1). In panel **(B)**, the grey hatched region represents $EMG_{total}$, which serves as the denominator in the Falconer-Winter CCI (Eqn. 2).

characterized by a single data point (i.e., peak EMG), here we express the $CCI_{SR}$ based on the integrated area of two EMG signals:

$$CCI_{SR} = \frac{\int EMG_{low}}{\int EMG_{high}}$$

(1)

The simple ratio also does not consider the general amplitude of activation or provide an indication of when or how co-contraction varies over time, which may lead to inaccurate or inflated estimates of co-contraction. The simple ratio is therefore best suited for tasks with relatively invariant EMG amplitudes, such as isometric contractions, rather than more dynamic movements, especially if the antagonist is unclear.

### 2.2. The Falconer and Winter Co-Contraction Index ($CCI_{FW}$)

Falconer and Winter [1] quantified co-contraction between two antagonist ankle muscles during gait using the ratio of the overlapping area between $EMG_1$ and $EMG_2$ (i.e., the integral of $EMG_{low}$) to the area of total EMG activity (i.e., integral of the sum of $EMG_1$ and $EMG_2$) (Eq. 2; Fig 1B). Multiplying the numerator ($EMG_{low}$) by two allows values of $CCI_{FW}$ to vary between 0 and 1. Although the original formula designated $EMG_{high}$ as the agonist and $EMG_{low}$ as the antagonist, explicitly defining the antagonist is not necessary when using $CCI_{FW}$.

$$CCI_{FW} = \frac{2 \int EMG_{low}(t)dt}{\int (EMG_1 + EMG_2)(t)dt}$$

(2)

The Falconer-Winter CCI is among the most widely cited co-contraction indices and has been used to study gait-related co-contraction in older adults [37,38], individuals with cerebral palsy [39], and stroke survivors [28]. Like the simple ratio, the Falconer-Winter CCI does not consider activation amplitude and was not originally intended to indicate when or how co-contraction varies over time. These characteristics suggest it is best applied to tasks with relatively constant EMG amplitudes or to short, quasi-stationary analysis windows. Historically reported as a single value integrated over brief time windows [1,38–40], the $CCI_{FW}$ has recently been computed sample-by-sample to produce a time-series [30].

## 2.3. The Thoroughman and Shadmehr Co-Contraction Index (CCI<sub>TS</sub>)

To identify EMG correlates of sensorimotor adaptation, Thoroughman and Shadmehr [33] quantified muscle co-contraction using two EMG-derived time-series: the "wasted contraction" (Eq. 3) and the "effective contraction" (Eq. 4). At each time point, the smaller of the two EMG signals (i.e., $EMG_{low}$) is labeled the "wasted contraction" because it is considered to be cancelled by the larger, opposing activation signal (i.e., $EMG_{high}$).

$$C_{wasted}(t) = EMG_{low}(t) \tag{3}$$

Subtracting the "wasted contraction" from the $EMG_{high}$ yields the "effective contraction" time-series.

$$C_{effective}(t) = EMG_{high}(t) - EMG_{low}(t) \tag{4}$$

The level of co-contraction can be quantified as the "wasted contraction" divided by the maximum of the "effective contraction", and either left as a time series for analysis [41] or expressed as a single index by taking the mean (Eq. 5; Fig 2) [28,33].

$$CCI_{TS} = mean\left(\frac{C_{wasted}(t)}{max(C_{effective}(t))}\right) \tag{5}$$

Notably, the Thoroughman-Shadmehr CCI has no upper bound. When EMG₁ and EMG₂ have very similar amplitudes the "effective contraction" (i.e., $EMG_{high}$—$EMG_{low}$) becomes very small. As a result, dividing by the max effective contraction—a very small number—produces co-contraction values that can become arbitrarily large, approaching infinity. If the two EMG signals are identical, the maximum "effective contraction" is zero and the CCI becomes undefined. The Thoroughman-Shadmehr CCI has been used primarily to study co-contraction in the upper [7,33,41,42] rather than lower limb [28]. In some studies the wasted contraction is not scaled by the maximum effective contraction. The CCI is instead expressed as the average wasted contraction [7,42], which is equivalent to the Unnithan-Frost CCI described below.

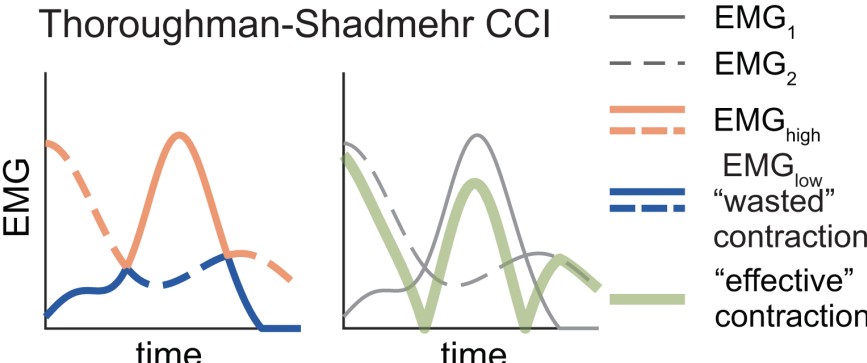

**Fig 2. Visualizing the Thoroughman-Shadmehr co-contraction index (CCI).** In both panels, solid and dashed lines represent EMG₁ and EMG₂, respectively. At each time point, the orange line corresponds to the EMG<sub>high</sub> signal, while the blue line corresponds to the EMG<sub>low</sub> signal (i.e., the "wasted" contraction). The solid green line represents EMG<sub>high</sub>— EMG<sub>low</sub>, termed the "effective" contraction. The "wasted" and the "effective" contraction are then used to calculate the value of the Thoroughman-Shadmehr CCI (Eqn. 5).

## 2.4 The Unnithan and Frost Co-Contraction Index (CCI$_{UF}$)

Unnithan et al., and Frost et al. [34,43] quantified co-contraction between agonist-antagonist muscle pairs in typically developing children and those with cerebral palsy using the ratio of the overlapping area between $EMG_1$ and $EMG_2$ (i.e., the integral of $EMG_{low}$) to the total number of data points ($N$) (Eq. 6; Fig 3A).

$$CCI_{UF} = \frac{\int EMG_{low}}{N}$$

(6)

The Unnithan-Frost CCI is equivalent to the mean of $EMG_{low}$ and can therefore be thought of as the average amplitude of the overlap between two antagonist EMG signals (see Fig 3A). CCI$_{UF}$ and CCI$_{TS}$ are equivalent if there is at least one time point where the difference between $EMG_{high}$ and $EMG_{low}$ is exactly one. In that scenario, the maximum "effective contraction"—which serves as the denominator of the Thoroughman-Shadmehr index (Eq. 5)—is equal to 1, and CCI$_{TS}$ becomes the same as the mean of $EMG_{low}$ (i.e., CCI$_{UF}$).

## 2.5 The Rudolph Co-Contraction Index (CCI$_R$)

While investigating knee stability after ACL injury, Rudolph et al. [32] quantified co-contraction between an agonist-antagonist muscle pair by computing the ratio of the less active muscle's amplitude (i.e., $EMG_{low}$) to the more active muscle's amplitude (i.e., $EMG_{high}$), multiplied by the sum of both muscles' activity at each time point $t$ during the time period of interest (Eq. 7; Fig 3B).

$$CCI_R(t) = \frac{EMG_{low}}{EMG_{high}} \left( EMG_1 + EMG_2 \right)$$

(7)

Unlike other CCIs, the Rudolph index is traditionally calculated as a time-series. By treating co-contraction as a function of time—rather than collapsing it into a single integrated, summed, or averaged value—this approach can reveal meaningful information about the magnitude and timing of peak co-contraction. This is particularly valuable for longer duration movements (e.g., over the course of a full gait cycle rather than during a single gait event or period), or when the timing of peak co-contraction is unknown. A single value for $CCI_R$ can be obtained by integrating the time-series, as originally proposed by Rudolph et al. [32]. Some researchers have instead used the sum [44] or average [45] of the $CCI_R$ time-series. In alignment with the original method, we report the integral throughout this paper. We discourage the use of the sum, as it

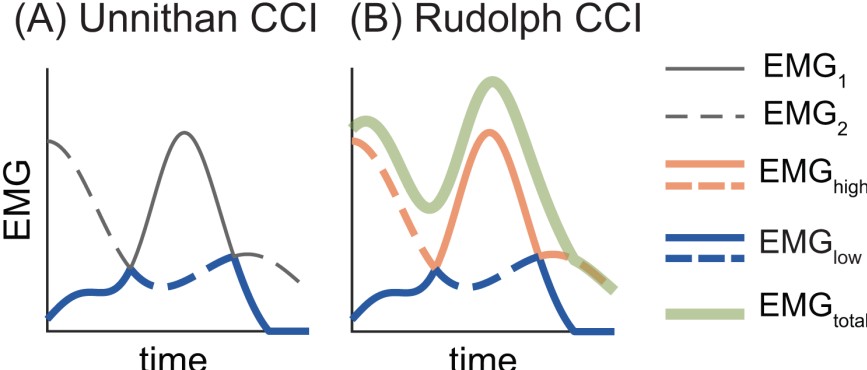

**Fig 3. Visualizing the Unnithan (A) and Rudolph (B) co-contraction indices (CCIs).** In both panels, solid lines represent EMG$_1$ and dashed lines represent EMG$_2$. At each time point, the blue and orange lines correspond to the EMG$_{low}$ and EMG$_{high}$ signals. In panel (B) the solid green line represents EMG$_{total}$ (i.e., EMG$_1$+EMG$_2$).

produces co-contraction estimates that are both artificially large [44] and dependent on the number of data points, thereby limiting comparisons across subjects or studies.

## 2.6 The temporal co-contraction index (CCI$_T$)

All of the CCIs described above incorporate both the amplitude and duration of the agonist and antagonist EMG signals. Co-contraction indices can also be derived solely from the temporal features of two EMG signals [46]. In a study of ankle muscle co-contraction after stroke, Lamontagne et al., [46] quantified temporal co-contraction between an agonist-antagonist muscle pair as the percentage of time that both muscles were active—that is, the ratio of their overlap time to the total observation time (Eq. 8; Fig 4). Here, $t_{overlap}$ represents the number of time points when both muscles are simultaneously active, and $t_{total}$ is the total number of time points within the period of interest.

$$CCI_T = \frac{t_{overlap}}{t_{total}} * 100\%$$

(8)

The magnitude of a temporal CCI can vary from 0% to 100% of the observation period. A temporal CCI may be appropriate when only the duration of simultaneous activation is of interest, or when working with populations in which EMG amplitude is unreliable. Unlike each of the other indices described above, $CCI_T$ is sensitive to the methods chosen to identify EMG onset and offset [47,48]. Other indices may therefore be preferable to $CCI_T$ if EMG onsets and offsets are difficult to identify. A further limitation of temporal CCIs is that they do not account for activation magnitude—co-contraction is treated the same whether both muscles are minimally or maximally active. $CCI_T$ is therefore insensitive to differences in EMG intensity.

## 3. A proposed co-contraction index classification scheme

We propose that the six CCIs reviewed above can be grouped into three categories: *shape-based*, *amplitude-driven*, and *temporal CCIs*. This classification scheme is based on the mathematical formula and general behavior of each index, as illustrated by 3D surface plots depicting all possible co-contraction values between two EMG signals varying from 0 to 1 (Fig 5) [49].

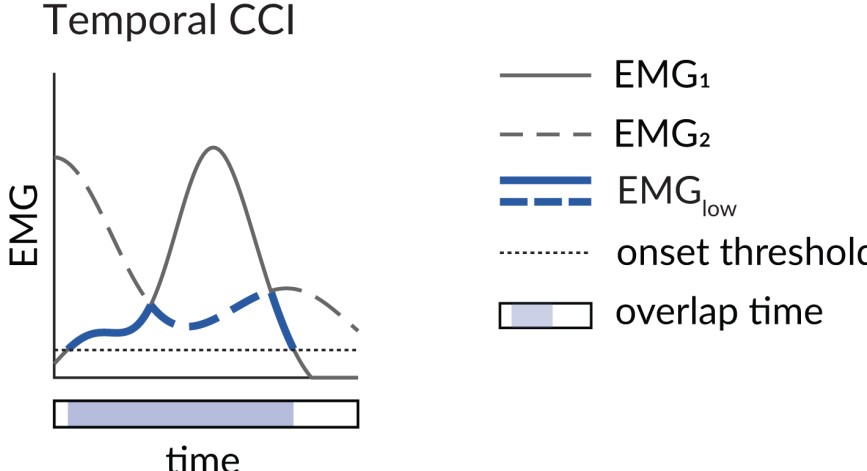

**Fig 4. Visualizing the temporal co-contraction index (CCI).** Solid lines represent EMG$_1$ and dashed lines represent EMG$_2$. Blue lines denote the low EMG signal at each time point. The shaded bar below the x-axis indicates the time points during which EMG$_1$ and EMG$_2$ overlap relative to the total time (black outline).

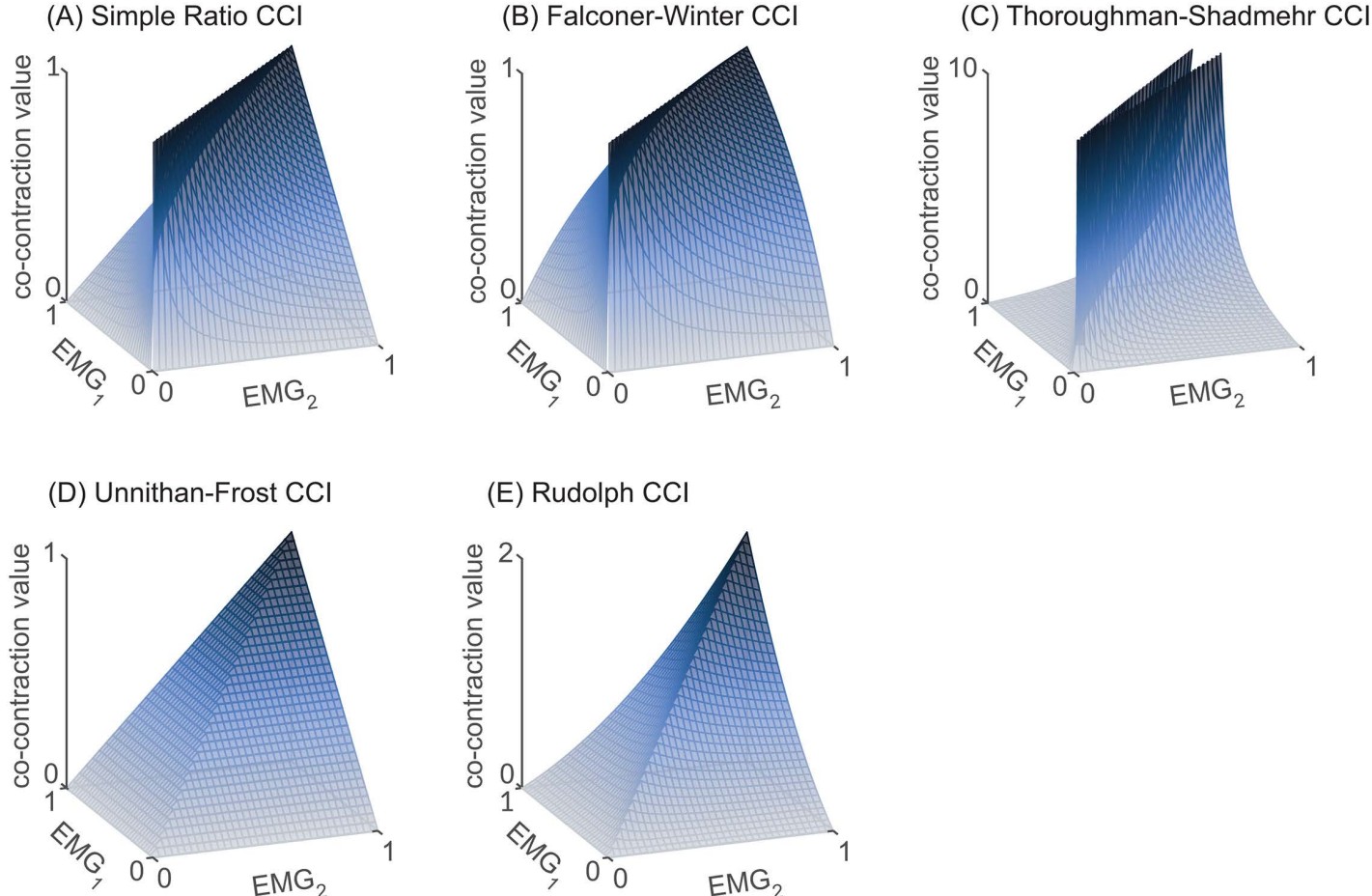

**Fig 5. 3D surface plots depicting all possible values of each co-contraction index (CCI) for any combination of EMG$_1$ and EMG$_2$.** Note the differences in shape, and thus behavior, of the CCIs. Panels A-C exhibit a dark blue "ridge" of high co-contraction whereas panels D and E possess a "sharp point" or peak of high co-contraction. The Simple Ratio (A), Falconer-Winter (B), and Thoroughman-Shadmehr (C) CCIs all return high co-contraction values when the two EMG signals have a similar value or shape, even when their amplitude is low (hence the "ridge"). These three indices are therefore termed shape-based CCIs. In contrast, the Unnithan-Frost and Rudolph CCIs both require high amplitude to produce high levels of co-contraction (hence the distinct peak-like feature in their surface plots). Because these two CCIs are primarily influenced by signal amplitude and require high activation in both muscles to produce large co-contraction values, they are termed amplitude-based CCIs. The temporal CCI is not included in the 3D surface plots because it does not vary with EMG amplitude. As CCI formulas do not all have the same maximum, y-axis limits in the surface plots vary.

A central difference among the six CCIs emerges from the shapes of their 3D surface plots (Fig 5). The simple ratio [36], Falconer-Winter [1], and Thoroughman-Shadmehr [33] CCIs all display a prominent "ridge" along the diagonal where EMG$_1$ and EMG$_2$ are similar (shown in dark blue, Fig 5A-C). This ridge indicates that co-contraction estimates from these indices are primarily influenced by the similarity in the shape of the two EMG signals, producing higher co-contraction values when the signals are similar, regardless of their amplitude (Fig 5A-C). Because these three indices share this characteristic "ridge" and are all computed as the ratio of the less active muscle (i.e., EMG$_{low}$) to another quantity (e.g., total EMG activity), we propose grouping them into a single category collectively referred to as *shape-based* CCIs. Except for the Thoroughman-Shadmehr index [33], which ranges from zero to infinity, shape-based CCIs generally range from 0 to 1. A value of 0 indicates no co-contraction and occurs when EMG$_{low}$ is zero, while a value of 1 typically reflects complete

co-contraction—when the two EMG signals are identical, though not necessarily high in amplitude. Shape-based CCIs can also be expressed as percentages when multiplied by 100.

While all shape-based CCIs share the characteristic "ridge", they also exhibit distinct features. Fig 6 illustrates the relationship between the three shape-based CCIs and $EMG_{low}$, assuming a fixed value for $EMG_{high}$, (i.e., taking a "slice" through each surface plot). The simple ratio CCI increases linearly with $EMG_{low}$, reflecting a steady increase in co-contraction as the amplitude of the two EMG signals become more similar. $CCI_{FW}$ exhibits a more convex profile, whereas the Thoroughman-Shadmehr CCI exhibits an asymptotic response, with values rising rapidly from zero towards infinity (Fig 6). These differences demonstrate that, even within the same CCI category, identical inputs can yield different co-contraction values. Here $EMG_{high}$ was set equal to 1, but the general patterns observed in Fig 6 hold for different values of $EMG_{high}$.

In contrast to shape-based CCIs, the surface plots of the Unnithan-Frost [34,43] and Rudolph [32] indices each display a single peak when both EMG signals reach their maximum amplitude (Fig 5D-E). This pattern illustrates that these indices are primarily influenced by signal amplitude and require high activation in both muscles to produce large co-contraction values. Based on this shared behavior, we propose grouping these two indices into a single category: *amplitude-driven* CCIs. Although amplitude-driven CCIs have no theoretical upper bound, when EMG signals are amplitude normalized to range from zero to one, the maximum possible values of $CCI_{UF}$ and $CCI_R$ are 1 and 2, respectively. Amplitude-driven CCIs yield values near zero when there is little or no antagonist muscle activity. In addition to their common dependence on amplitude, both amplitude-driven indices display distinct behavioral characteristics. For example, the Unnithan-Frost index increases linearly with increases in $EMG_{low}$ (Fig 5D), whereas the Rudolph index is more concave, reflecting a more "conservative" estimate of co-contraction (Fig 5E). Among all indices described here the Rudolph CCI is the most stringent: requiring high activation in both muscles before producing large co-contraction values.

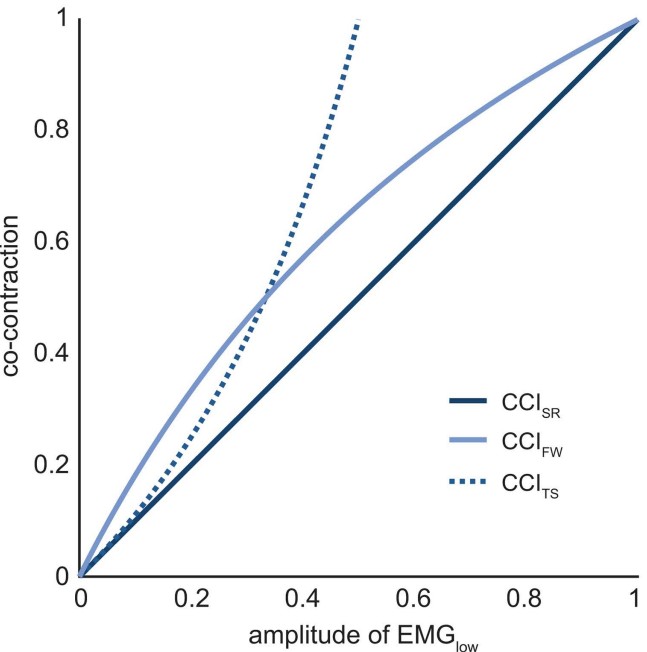

**Fig 6. Shape-based CCI values with $EMG_{high}$ fixed at 1.** Co-contraction values are presented for increasing values of $EMG_{low}$ while the value for $EMG_{high}$ is fixed at 1. Co-contraction values derived from the three shape-based indices increase at different rates as the amplitude of $EMG_{low}$ increases, suggesting that each index may yield distinct estimates of co-contraction, especially at moderate levels of $EMG_{low}$ (e.g., 0.5). Similar results are obtained if different values of $EMG_{high}$ are used. Abbreviations: $CCI_{SR}$: Simple Ratio; $CCI_{FW}$: Falconer-Winter; $CCI_{TS}$: Thoroughman-Shadmehr.

Because temporal CCIs are based on the simultaneous activation of two muscles regardless of their amplitude, they produce binary values (0 or 1) at each time point, making them poorly suited for visualization with a 3D surface plot.

Comparing shape-based, amplitude-driven, and temporal CCI behaviors raises a central question: *what constitutes co-contraction?* Must both the agonist and antagonist muscles be highly active, as emphasized by amplitude-driven CCIs? Or is it sufficient that their activation levels are similar, regardless of amplitude, as assumed by shape-based methods? Alternatively, can co-contraction be quantified independent of both signal amplitude and similarity, as is implied by temporal CCIs? These questions highlight the importance of recognizing that different methods for estimating co-contraction may yield divergent results under similar conditions. For instance, during sustained low-amplitude activation of an agonist-antagonist muscle pair, shape-based CCIs would identify high co-contraction, while amplitude-driven CCIs would not. One could argue that amplitude-driven methods may underestimate co-contraction in such scenarios, potentially over-looking small but meaningful contributions to joint stabilization or movement inefficiency. Ultimately, shape-based, amplitude-driven, and temporal CCIs likely capture distinct, rather than redundant, aspects of co-contraction—each offer-ing unique insights into human movement—an idea explored in greater detail below.

## 4. Do the groups hold up? Testing the proposed CCI classification scheme

The initial validity of the proposed CCI classification scheme was evaluated using correlation analyses of CCIs calcu-lated using synthetic EMG data. Stronger correlations among CCIs within rather than between groups (i.e., shape-based, amplitude-driven, or temporal) would provide initial evidence supporting the proposed classification. Synthetic data were used to enable a more precise evaluation of CCI behavior than would be possible with experimental EMG data. Three thousand pairs of synthetic EMG signals—each ranging in amplitude from 0 up to 1—were generated to represent ampli-tude normalized EMG data from two muscles. The synthetic EMG data was generated as sinusoids (N = 1000), first-(N = 1000), and second-order (N = 1000) polynomials, in order to capture a large range of signals with varying shapes. Polynomials were created using randomly generated coefficients between −1 and 1 and intercepts between 0 and 1 for each muscle. Sinusoids were generated in the form:

$$\begin{cases} EMG_1 = c_1 sin\left(c_2 * \left(t - c_3\right)\right) + c_4 \\ EMG_2 = c_5 cos\left(c_6 * \left(t - c_7\right)\right) + c_8 \end{cases}$$

(9)

where $c_2$ and $c_6$ are random numbers between 0 and 10, and all other $c_i$ are random numbers between 0 and 1. All syn-thetic EMG signals were rectified to eliminate negative values. Signals with a maximum amplitude greater than 1 were scaled to range from 0 to 1 (Fig 7). An activation threshold of 0.1 was used to define onset and offset for the temporal CCI only (for all other indices, nonzero EMG is considered active). While arbitrary, varying the threshold level was found to have minimal effect on the results.

Values for all six CCIs were calculated for each of the 3000 pairs of synthetic EMG signals. The relationship between co-contraction estimates from each of the six CCIs were visualized with scatterplots and quantified by calculating Chat-terjee's correlation coefficients [50]. Unlike Pearson's correlation coefficient—which only detects linear relationships—and Spearman's rho—which can capture nonlinear relationships for monotonic functions (i.e., always increasing or decreas-ing)—Chatterjee's coefficient can identify both nonlinear and nonmonotonic relationships (e.g., a parabola) [50,51], making it well suited for evaluating the strength of the relationships between CCIs. Matlab scripts used to generate and evaluate the synthetic EMG data are available on GitHub (DOI: 10.5281/zenodo.20058872).

Except for correlations involving the temporal CCI, Chatterjee correlation coefficients were positive and greater than 0.4 (Fig 8). Consistent with our hypothesis, the strongest correlations occurred within each of the proposed CCI groups, i.e., >0.7 for shape-based and > 0.9 for amplitude-driven). This pattern lends support to the proposed classifi-cation scheme. However, correlations among all five shape-based and amplitude-driven CCIs were consistently > 0.5,

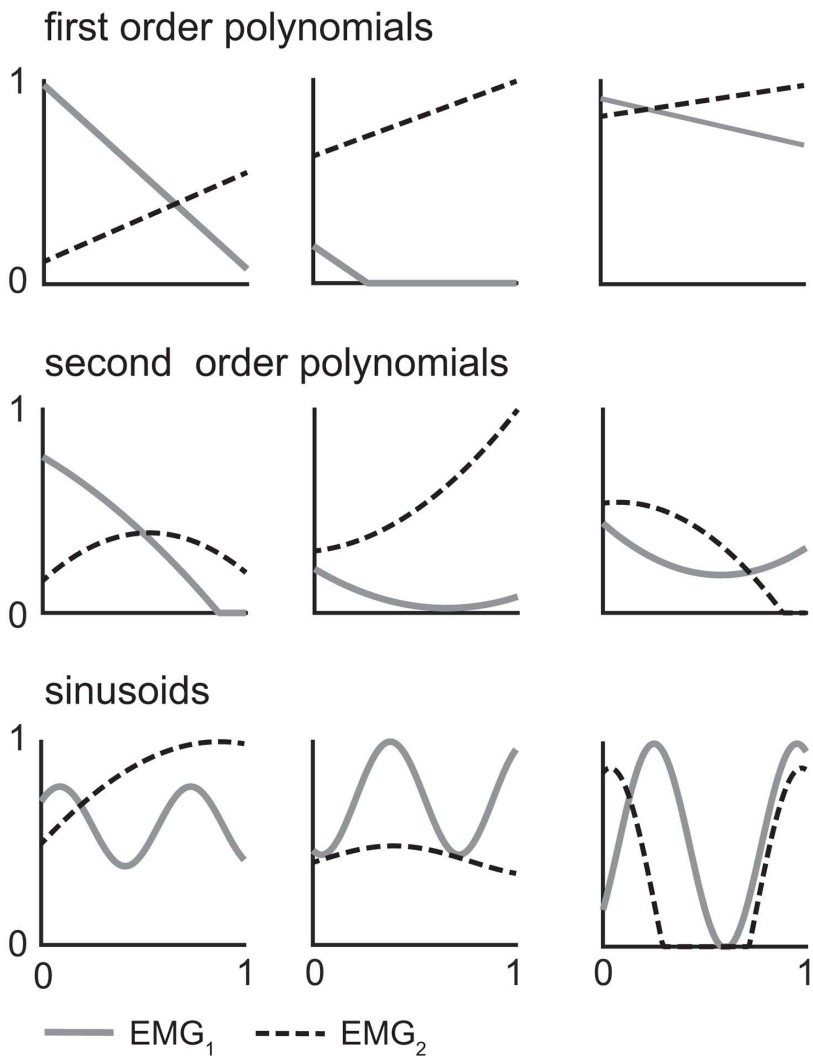

**Fig 7. Samples of randomly generated synthetic EMG signals used to quantify correlations between co-contraction indices.**

suggesting some degree of similarity between shape-based and amplitude-driven indices. Although shape-based and amplitude-driven CCIs produce different absolute values, their moderate correlations suggest that relative trends may be comparable across studies. However, this holds only when the underlying EMG features (i.e., amplitude, shape)—and nature of co-contraction being studied—are similar across studies.

To this point, we have characterized the behavior and relationships among commonly cited CCIs and grouped each index into one of three categories: (i) shape-based CCIs, which quantify similarity in shape between EMG signals; (ii) amplitude-driven CCIs, which capture periods of simultaneously high activation; and (iii) temporal CCIs, which quantify the duration of simultaneous activation. Within-category correlations were strong and behaviors similar, whereas between-category correlations were weaker, suggesting that these indices capture complimentary, rather than redundant aspects of co-contraction. Next we present three practical, data-driven recommendations that address common methodological and conceptual challenges in selecting, calculating, and interpretating CCIs.

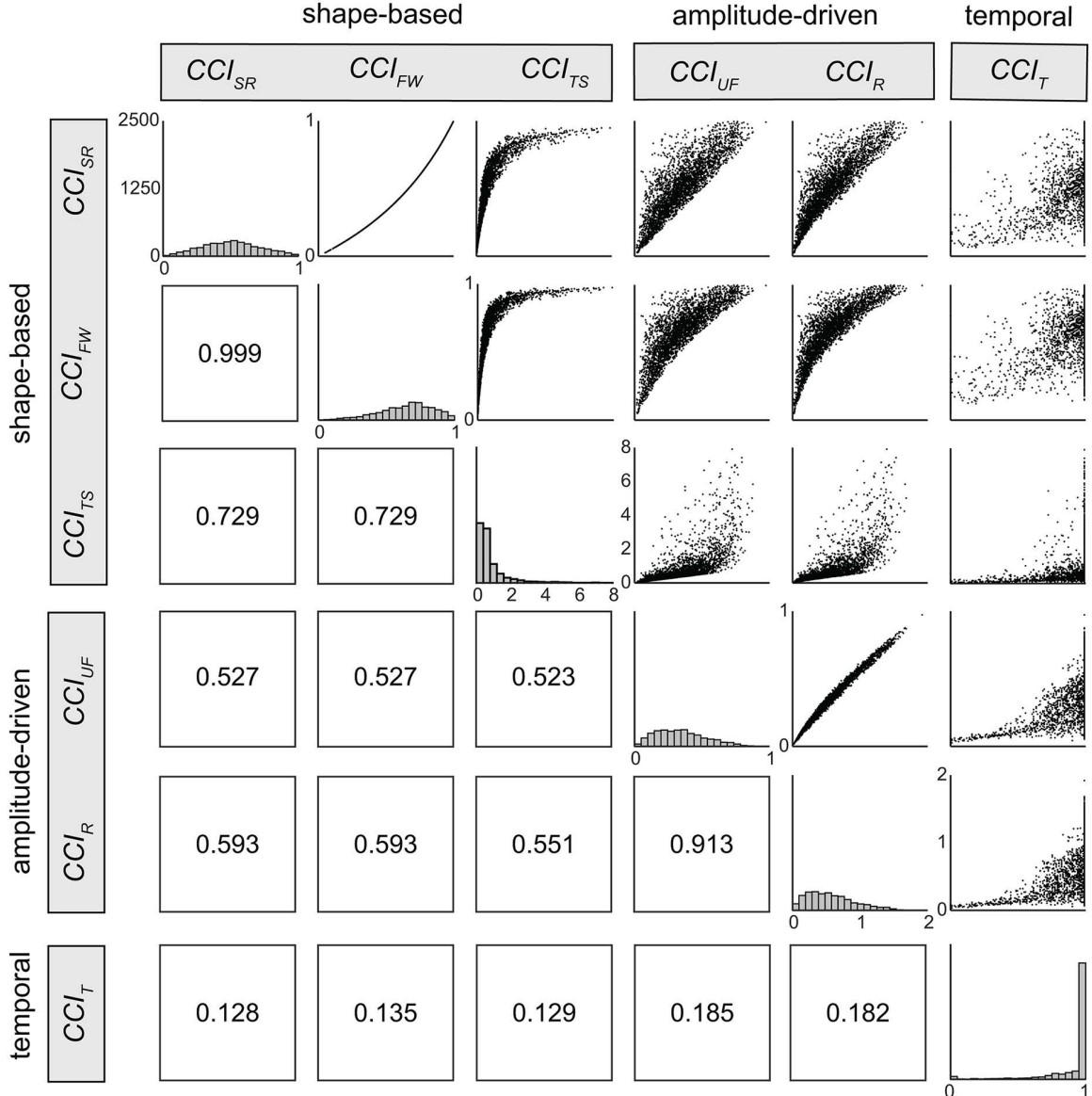

**Fig 8. Pairwise correlations among six CCIs, derived from randomly generated pairs of synthetic EMG data, were quantified using Chatter-jee's correlation coefficient and visualized with scatterplots.** Axis limits differ across scatterplots but remain consistent within each row and column. Histograms along the diagonal illustrate the distribution of co-contraction values for each index, with co-contraction values on the x-axis and incidence on the y-axis (0-2500). Correlations were strongest among CCIs within the same group (i.e., shape-based or amplitude-based indices), supporting the proposed classification scheme. With the exception of the temporal CCI, moderate correlations were also observed between CCIs from different groups, a pattern which suggests that relative trends but not absolute values may be cautiously compared across studies that use indices from different groups. Abbreviations: $CCI_{SR}$: Simple Ratio; $CCI_{FW}$: Falconer-Winter; $CCI_{TS}$: Thoroughman-Shadmehr; $CCI_{UF}$: Unnithan-Frost; $CCI_R$: Rudolph; $CCI_T$: Temporal.

## 5. Recommendations for the selection, calculation, and interpretation of co-contraction indices

To inform each of the recommendations below, synthetic EMG data were generated to create controlled and well-defined conditions that would be difficult to achieve with experimental data. Distinct from the correlation analysis, these synthetic signals were specifically designed to reflect common features of experimental EMG—such as low-level activation and

varying degrees of overlap—that could be used to test each index's response to specific signal characteristics. Eight pairs of synthetic EMG signals were generated using linear or piecewise linear functions, each ranging in value from 0 to 1, to reflect EMG data normalized to a within-task maximum and evaluated over 100 time points. The Matlab scripts used to generate and analyze these signals are provided on GitHub (DOI: 10.5281/zenodo.20058872).

<u>Proposal 1</u>: Different amplitude normalization techniques alter the magnitude and interpretation of co-contraction indices

Amplitude normalization of EMG signals to a maximum reference value enables meaningful comparisons across muscles, participants, and measurement sessions or conditions [21]. The choice of amplitude normalization technique depends on the intended analysis and interpretation [21,52]. While most studies apply some form of amplitude normalization prior to calculating co-contraction values [26–28,53], the effects of different normalization techniques on co-contraction estimates remain unclear. Several studies have investigated how select normalization techniques influence co-contraction, yielding mixed results [28,54,55]. One study found that in the paretic limb of individuals with chronic stroke, co-contraction estimates between the soleus and tibialis anterior differed significantly between maximum M-wave and within-task normalization techniques across multiple shape-based CCIs [28]. Most evidence to date, however, suggests that amplitude normalization has little to no effect on co-contraction values [28,54,55]. These conclusions are based exclusively on shape-based CCIs [28,54,55], have been made without reference to non-normalized data [21], and in some cases, rely on EMG collected from neurologically impaired adults (e.g., post-stroke) [28]. An evaluation that incorporates amplitude-driven CCIs and comparisons to non-normalized data is needed to clarify the impact of amplitude normalization on co-contraction estimates.

To assess how different amplitude normalization techniques affect each CCI, three pairs of synthetic EMG signals, described above, were modified to reflect the peak amplitudes typically produced by three common amplitude normalization methods [21]: i) amplitudes ranging from 0 to 0.6, *consistent with normalization to a maximum voluntary contraction (MVC) or M-wave*; ii) amplitudes ranging from 0 to 1, *representing normalization to a within-task maximum*; and iii) amplitudes ranging from 0 to 1.4, *simulating non-normalized* EMG data. These differences in signal amplitude were then evaluated for their effect on co-contraction estimates calculated using the indices reviewed above.

EMG amplitude normalization techniques appear to affect shape-based and amplitude-driven CCIs differently (Fig 9). Amplitude-driven CCIs (e.g., $CCI_R$, $CCI_{UF}$) were most sensitive to normalization methods that altered *both* the amplitude of $EMG_{low}$ *and* the total EMG activity (Fig 9A & C). Compared to non-normalized EMG data (dotted line, Fig 9A), decreases in the amplitude of $EMG_2$ following within-task (dot-dashed line) and MVC-based amplitude normalization (dashed line) reduced the overlap between $EMG_1$ and $EMG_2$ while also slightly reducing the total EMG activity. The decrease in $EMG_2$, which serves as $EMG_{low}$ for the first half to two-thirds of Fig 9A, reduces the numerator of amplitude-driven CCIs (see Fig 3), thereby decreasing their overall magnitude (Fig 9C). Although the numerator of shape-based CCIs (e.g., Falconer-Winter, Fig 1) is slightly influenced by the decrease in $EMG_{low}$, a simultaneous reduction in total EMG activity—the denominator in shaped-based CCIs - offset this effect. As a result, the ratio of $EMG_{low}$ to $EMG_{high}$, and thus shape-based CCIs would remain relatively unchanged if both the amplitude of $EMG_{low}$ *and* the total EMG activity are altered by amplitude normalization.

In contrast, when only *one* of either total EMG activity or $EMG_{low}$ is altered by amplitude normalization, the ratio of $EMG_{low}$ to $EMG_{high}$ changes, producing a large increase in the magnitude of shaped-based CCIs (Fig 9B & D). For example, in Fig 9B, a decrease in the amplitude of $EMG_2$, which serves as $EMG_{high}$ throughout, due to within-task (dot-dashed line) and MVC-based normalization (dashed line) reduces the total EMG activity without affecting the overlap between $EMG_1$ and $EMG_2$. This selective reduction in the denominator of shape-based CCIs (e.g., $CCI_{SR}$, $CCI_{FW}$, $CCI_{TS}$; Figs 1 and 2) increases the co-contraction values (Fig 9D). Amplitude-driven CCIs are largely unaffected in such cases because they are more sensitive to changes in signal overlap, which depends on the amplitude of $EMG_{low}$ (Fig 9D).

Amplitude normalization is largely irrelevant to temporal CCIs (e.g., Lamontagne) [46], which are based on EMG onset and offset times rather than signal amplitude or shape. In theory, normalizing EMG data to an extremely low reference

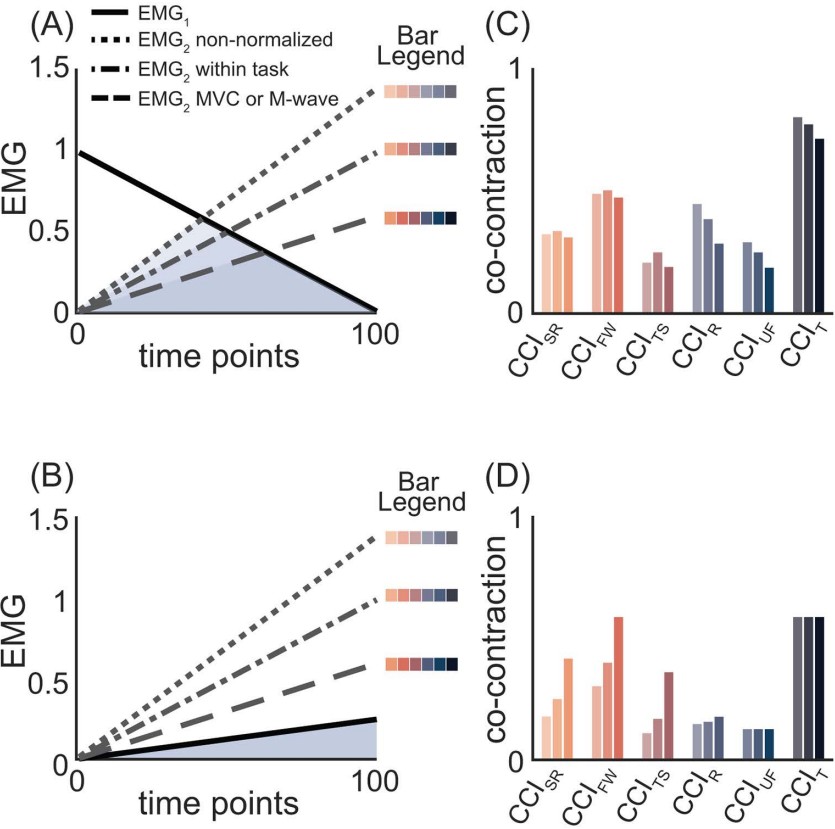

**Fig 9. Amplitude normalization influences each CCI in distinct ways.** In panels A and B, the amplitude of EMG₁ (solid black lines) is fixed, while that of EMG₂ varies to represent three amplitude normalization conditions: non-normalized (dotted line, maximum amplitude 1.4), within-task normalization (dash-dotted line, maximum amplitude 1), and MVC normalization (dashed line, maximum amplitude 0.6). Panels C and D show bar plots of co-contraction values for all six CCIs, calculated from the three EMG₁ and EMG₂ combinations in panels A and **B.** Amplitude-driven CCIs (i.e., Rudolph and Unnithan-Frost) are most affected when amplitude normalization reduces *both* EMG_low and total EMG (panel **C**). In contrast, shaped-based CCIs, like Falconer-Winter, are most sensitive when normalization alters *only one of* EMG_low and total EMG (panel **D**). Abbreviations: CCI_SR: simple ratio; CCI_FW: Falconer-Winter; CCI_TS: Thoroughman-Shadmehr; CCI_R: Rudolph; CCI_UF: Unnithan-Frost; CCI_T: temporal.

value could increase the signal-to-noise ratio, complicate onset and offset detection, and potentially influence temporal CCI values. Such scenarios are unlikely in practice and the choice to normalize—or not—is generally not applicable to the calculation or interpretation of temporal CCIs.

These findings highlight the importance of considering the amplitude normalization method used [21], the type of CCI to which it is applied, and whether the method affects signal overlap (i.e., EMG_low), total EMG activity, or both. These factors determine whether—and in which direction—amplitude normalization alters estimates of muscle co-contraction.

**Recommendation**: Because the effect of amplitude normalization on co-contraction varies by index, studies should assess the sensitivity of their results to different normalization techniques when possible.

Proposal 2: Co-contraction estimates calculated from different indices are not directly comparable.

The wide range of behaviors (Fig 5) and different theoretical maximum values across CCIs (e.g., $CCI_{FW\_max} = 1$, $CCI_{R\_max} = 2$) suggest that estimates calculated from different CCIs should not be directly compared. Fig 10 illustrates how

co-contraction estimates vary across indices under three well defined conditions: none (A), maximal (B), and partial (C) overlap between two synthetic EMG signals.

In Fig 10A, EMG signals do not overlap, only one muscle is active at a time, and all CCIs correctly return values of zero. From this example we can infer that none of the six reviewed CCIs are prone to identifying co-contraction when

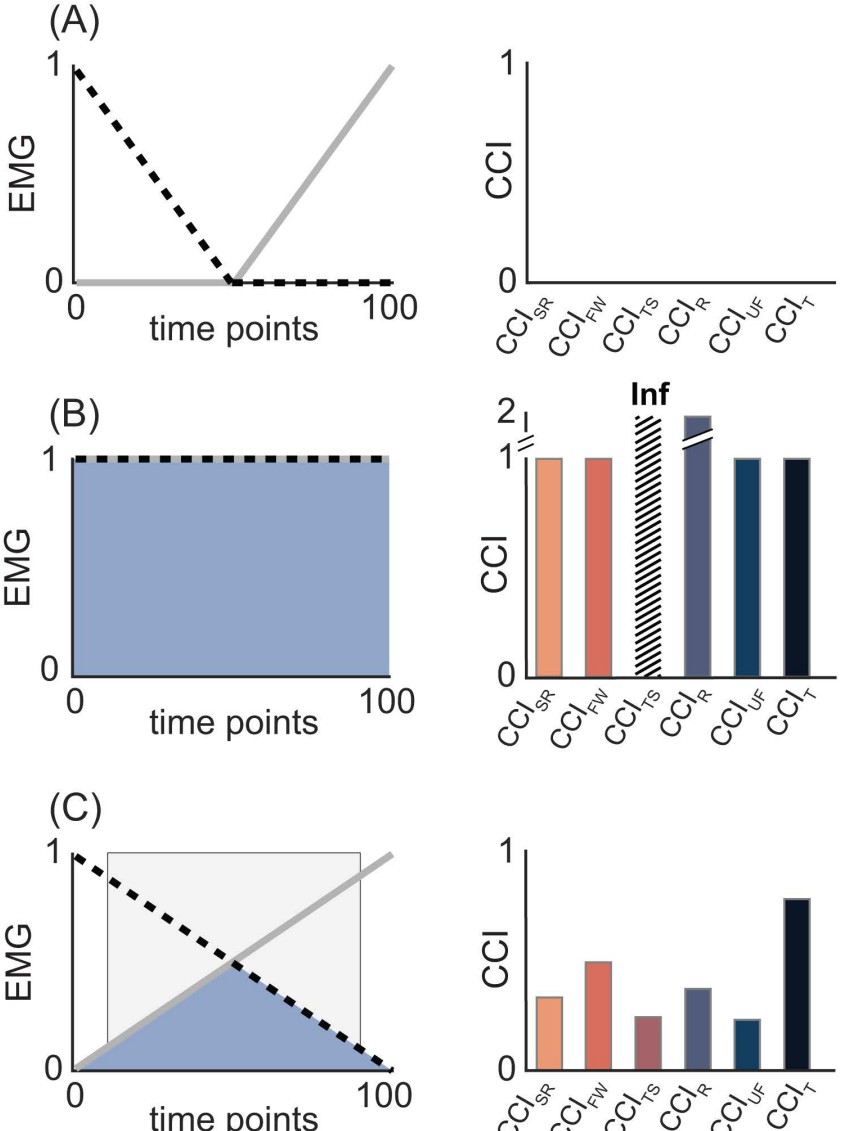

**Fig 10. Under identical conditions, different indices yield different co-contraction values.** Panels A-C illustrate three levels of overlap between two synthetic EMG signals plotted as a solid grey line and a dashed black line (A: none, B: complete, C: partial). Overlap is shown by the shaded blue regions, with corresponding bar graphs presenting the co-contraction values derived from each index for the three conditions. Light gray shading indicates periods of co-contraction identified by the temporal CCI. **(A)**: In the absence of overlap, all CCI correctly return a value of zero. **(B)**: Under complete overlap, all CCIs indicate "maximum" co-contraction, but the Thoroughman-Shadmehr (CCI$_{TS}$) and Rudolph (CCI$_R$) indices return higher numerical values. Because indices have different theoretical maximums, these higher values should not be misinterpreted as "more" co-contraction than the other indices, but rather as "full" co-contraction. **(C)**: Under partial overlap co-contraction values differ across indices. Subsequently, direct comparisons of indices' absolute values should be avoided. Abbreviations: CCI$_{SR}$: Simple Ratio; CCI$_{FW}$: Falconer-Winter; CCI$_{TS}$: Thoroughman-Shadmehr; CCI$_R$: Rudolph; CCI$_{UF}$: Unnithan-Frost; CCI$_T$: Temporal.

none exists. In Fig 10B, both muscles are maximally active throughout. Under these conditions the temporal and shape-based indices (e.g., $CCI_{SR}$, $CCI_{FW}$), along with $CCI_{UF}$, return a value of 1, but $CCI_R$ yields a value of 2 and $CCI_{TS}$ is undefined. In the absence of an awareness for the different theoretical maximum values of each CCI, comparing $CCI_{FW}$ and $CCI_R$ in this scenario would falsely suggest that a study using $CCI_R$ observed "more co-contraction" than one using $CCI_{FW}$, when the difference is purely mathematical—not physiological. Between these two extremes, when $EMG_1$ and $EMG_2$ partially overlap (Fig 10C), the calculated co-contraction values vary across indices (e.g., $CCI_{SR} = 0.33$; $CCI_{FW} = 0.50$; $CCI_{TS} = 0.25$; $CCI_{UF} = 0.25$; $CCI_R = 0.39$; $CCI_T = 0.77$). Consistent with prior studies [18,28] these results show substantial variation in co-contraction values across CCIs, even when computed from identical EMG signals. Such differences hinder cross-study comparisons when different indices are used [56], complicating the interpretation of systematic reviews [25,27,56].

**Recommendation**: Avoid direct comparisons of co-contraction values derived from different indices. Instead, evaluate relative trends with respect to each index's theoretical maximum to ensure fair, meaningful comparisons across CCIs.

Proposal 3: The choice of co-contraction index should be aligned with hypothesized change (or difference) in EMG signals.

As discussed above and illustrated in Fig 5, each category of CCIs are sensitive to different characteristics of EMG signals and arguably represent different concepts of co-contraction. CCIs should therefore be selected based on the EMG features expected to differ across tasks or groups that are relevant to the research question (i.e., amplitude or shape). Amplitude-driven CCIs reflect high simultaneous activation; they are sensitive to changes in EMG signal amplitude and will therefore detect condition-dependent differences in co-contraction when amplitude changes occur without meaningful changes in EMG shape. In contrast, shape-based CCIs reflect waveform similarity; they are sensitive to signal shape and largely insensitive to amplitude differences. Selecting a CCI that does not reflect the hypothesized concept of co-contraction could limit detection of experimental differences.

As an example consider a study examining whether co-contraction differs between walking on level and uneven terrain. Prior work suggests that walking on uneven terrain increases EMG amplitude in both agonist and antagonist muscles while producing little change in signal shape [57]. In this case, differences in co-contraction are most likely to be detected with an amplitude-driven CCI that captures simultaneous increases in EMG magnitude. Fig 11 illustrates this point. The shape-based CCI (e.g., $CCI_{FW}$) yields similar values for level terrain (Fig 11A) and uneven terrain (Fig 11B), consistent with minimal waveform changes. In contrast, the higher EMG amplitude (and longer activation) on uneven terrain increases the amplitude-driven and temporal CCIs (e.g., $CCI_R$, $CCI_T$). Choosing a shape-based CCI in this context could mask meaningful differences between walking conditions.

This example is not intended to represent all possible situations; rather, it illustrates that no single CCI captures all signal features that influence co-contraction estimates. Instead, each index emphasizes different EMG features (i.e., amplitude or shape), and their suitability depends on the task- and condition-specific determinants of co-contraction. The theoretical sensitivity of shape-based and amplitude-driven CCIs to different features of EMG was demonstrated experimentally by Harrington et al., (2025). In both typically developing children and those with cerebral palsy, co-contraction was higher during fast versus slow walking, and during overground versus aquatic treadmill walking—but only when calculated using amplitude-driven CCIs (i.e., $CCI_{UF}$ and $CCI_R$). Shape-based CCIs (e.g., $CCI_{FW}$) did not capture these differences [30]. That only amplitude-driven, not shape-based, CCIs detected these task-specific difference in co-contraction is attributable to the observed differences in EMG amplitude and the absence of any notable differences in signal shape between experimental conditions [30]. These results support both our proposed classification scheme and the broader recommendation to select CCIs based on expected or observed differences in EMG signal amplitude and/or shape.

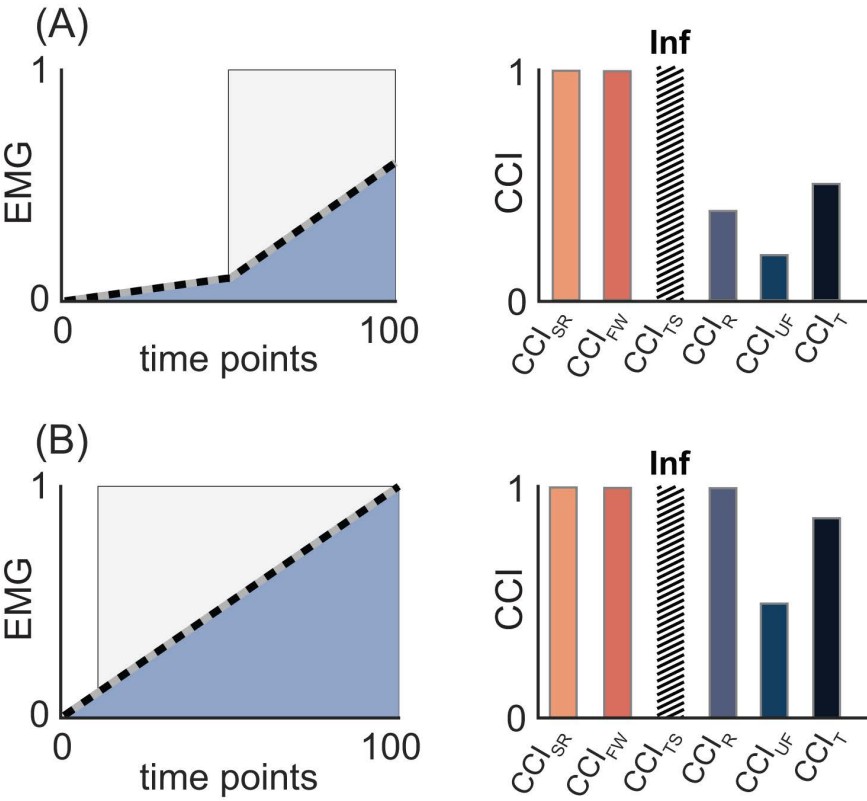

**Fig 11. Align the choice of CCI with the EMG feature expected to change or differ.** Synthetic EMG signals are plotted as solid gray and dashed black lines, with overlapping regions shaded blue. Light gray shading indicates periods of co-contraction identified by the temporal CCI. Corresponding bar graphs display the co-contraction values calculated from each index under each condition. Because of their sensitivity to signal similarity, shape-based CCIs (i.e., $CCI_{SR}$, $CCI_{FW}$, $CCI_{TS}$)—but not amplitude-driven (e.g., $CCI_{R}$, $CCI_{UF}$) or temporal CCIs—return identical values in panels A and B, indicating "complete" co-contraction despite differences in signal amplitude. In contrast, amplitude-driven, but not shape-based CCIs reveal their sensitivity to signal amplitude by producing smaller co-contraction values in panel A than in **B.** Variations in co-contraction due to signal similarity are best detected with shape-based CCIs, whereas differences in co-contraction arising from changes in signal amplitude are better captured with amplitude-based CCIs. Abbreviations: $CCI_{SR}$: simple ratio; $CCI_{FW}$: Falconer-Winter; $CCI_{TS}$: Thoroughman-Shadmehr; $CCI_{R}$: Rudolph; $CCI_{UF}$: Unnithan-Frost; $CCI_{T}$: temporal.

**Recommendation**: Match the co-contraction index to the co-contraction concept and thus EMG feature of interest—use a shape-based index (e.g., Falconer-Winter) when signal similarity is prioritized, and an amplitude-driven index (e.g., Rudolph) when differences in signal magnitude and duration are expected.

## 6. Final thoughts and next steps

We have shown that formulas for common co-contraction indices fall into one of three categories: i) *shape-based CCIs*, which define high co-contraction based on signal similarity regardless of amplitude, ii) *amplitude-driven CCIs*, in which simultaneous high activation in the agonist and antagonist muscles is required to yield high co-contraction, and iii) *temporal CCIs*, which defines co-contraction as the duration of simultaneous activation regardless of signal shape or amplitude. CCIs within each category—and, to a lesser extent, between categories—behave similarly. Due to differences in formula behavior and sensitivities to different features of EMG signals, we propose that researchers should: i) when possible, assess the sensitivity of their results to different amplitude normalization techniques, ii) assess relative trends rather than directly compare absolute co-contraction values derived from different indices, and iii) consider using a CCI sensitive to

expected changes or differences in EMG signals. The current results also emphasize the importance of initial descriptive and visual analyses of EMG data so that researchers can carefully select indices in accordance with the features of their EMG data and specific research question.

Here we focused on six commonly used CCIs. Many other indices have been proposed and may be better suited for some research questions. The time-varying multi-muscle co-activation function developed by Ranavolo et al., is an amplitude-driven CCI that can incorporate more than two muscles to provide a more global estimate of co-contraction [49,58]. The Vector Coding Technique [59], which determines if EMG signals are in-phase [31], and the correlation coefficient technique, which quantifies the direction and strength of a linear relationship between two muscle signals [29,60,61], may be better suited for questions related to shared input to agonist-antagonist muscle pairs involved in co-contraction.

Our analyses used synthetic EMG data to create controlled scenarios, enabling more precise evaluation of CCI behavior over a wider range of conditions than would be feasible with experimental EMG. An important next step is to test the robustness of these theoretical findings using experimental EMG from diverse tasks, muscle pairs, and clinical populations.

## Acknowledgments

We used the scientific colormaps oslo (for 3D surface plots) and lipari (for synthetic EMG) [62], which prevents visual distortion and is interpretable to readers with color-vision deficiencies [63].

## Author contributions

**Conceptualization:** Hannah D. Carey, Andrew Sawers.

**Data curation:** Hannah D. Carey.

**Formal analysis:** Hannah D. Carey.

**Funding acquisition:** Andrew Sawers.

**Investigation:** Hannah D. Carey.

**Methodology:** Hannah D. Carey, Friedl De Groote, Andrew Sawers.

**Project administration:** Andrew Sawers.

**Resources:** Andrew Sawers.

**Software:** Hannah D. Carey.

**Supervision:** Andrew Sawers.

**Validation:** Hannah D. Carey.

**Visualization:** Hannah D. Carey, Andrew Sawers.

**Writing – original draft:** Hannah D. Carey.

**Writing – review & editing:** Hannah D. Carey, Friedl De Groote, Andrew Sawers.

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
