## [Decision Letter · Decision Letter 0]

22 Apr 2026

PONE-D-26-05502A comparative analysis of co-contraction indices using synthetic EMG data: Implications for selection and interpretationPLOS One

Dear Dr. Sawers,

Thank you for submitting your manuscript to PLOS ONE. After careful consideration, we feel that it has merit but does not fully meet PLOS ONE’s publication criteria as it currently stands. Therefore, we invite you to submit a revised version of the manuscript that addresses the points raised during the review process.

**ACADEMIC EDITOR:** The manuscript presents a well-written and comprehensive review of muscle co-contraction indexes. However, please update it by addressing all the concerns raised by the reviewers.

A letter that responds to each point raised by the academic editor and reviewer(s). You should upload this letter as a separate file labeled 'Response to Reviewers'.

A marked-up copy of your manuscript that highlights changes made to the original version. You should upload this as a separate file labeled 'Revised Manuscript with Track Changes'.An unmarked version of your revised paper without tracked changes. You should upload this as a separate file labeled 'Manuscript'.

We look forward to receiving your revised manuscript.

Kind regards,

Andrea Tigrini, Ph.D.

Academic Editor

PLOS One

Journal Requirements:

The authors disclosed receipt of the following financial support for the research, authorship, and/or publication of this article: Research reported in this publication was supported by the Department of Defense

under Award No. HT9425-24-1-0212. The content is solely the responsibility of the authors and does not necessarily represent the official views of the Department of Defense.

Additional Editor Comments :

The manuscript presents a well-written and comprehensive review of muscle co-contraction indexes. However, please update it by addressing all the concerns raised by the reviewers.

Reviewers' comments:

Reviewer's Responses to Questions

**Comments to the Author**

1. Is the manuscript technically sound, and do the data support the conclusions?

Reviewer #1: Yes

2. Has the statistical analysis been performed appropriately and rigorously? 

Reviewer #1: N/A

3. Have the authors made all data underlying the findings in their manuscript fully available?

Reviewer #1: Yes

4. Is the manuscript presented in an intelligible fashion and written in standard English?

Reviewer #1: Yes

5. Review Comments to the Author

Reviewer #1: The manuscript is well written and the methodological framework is carefully developed. The comparison of multiple co-contraction indices is particularly interesting and represents a valuable methodological contribution. I only have a minor suggestion regarding Figure 9. While the normalization parameters are mentioned in the manuscript and the script used to generate the synthetic EMG signal is available in the GitHub repository, the intermediate steps that lead from those parameters to the plot shown in Figure 9 are not entirely clear from the text. It would therefore be helpful if the authors could briefly clarify the processing steps that transform the reported normalization parameters into the final signal representation shown in the figure.

6. PLOS authors have the option to publish the peer review history of their article (what does this mean?). If published, this will include your full peer review and any attached files.

Reviewer #1: No

---

## [Author Response · Author response to Decision Letter 1]

7 May 2026

Manuscript PONE-D-26-05502

Response to Reviewers

Dear Dr. Tigrini,

Thank you for the opportunity to address reviewer and editorial comments. We have provided the point-by-point response below. All tracked changes are noted as red underlined text for additions/changes, and red strikethrough text for deletions.

Editor’s Comments to Authors:

Response: We have reviewed the style template and revised the manuscript as appropriate.

Response: We have made all code available without restriction on GitHub, archived with Zenodo.

Response: All data are in the manuscript and/or supporting information files.

4. We note that the grant information you provided in the ‘Funding Information’ and ‘Financial Disclosure’ sections do not match. When you resubmit, please ensure that you provide the correct grant numbers for the awards you received for your study in the ‘Funding Information’ section.

Response: We have revised the grant information to match in the ‘Funding Information’ and ‘Financial Disclosure’ sections. The correct grant number is included in the ‘Funding Information’ section.

The authors disclosed receipt of the following financial support for the research, authorship, and/or publication of this article: Research reported in this publication was supported by the Department of Defense under Award No. HT9425-24-1-0212. The content is solely the responsibility of the authors and does not necessarily represent the official views of the Department of Defense.

Response: Thank you. The funder had not role. We have included the requested statement in our cover letter.

Response: N/A

Response: We have checked the reference list, which is complete. We have updated the formatting for consistency between items (i.e., abbreviation of journals). We have also added two references regarding the color palettes used in manuscript figures, reproduced below:

62. Crameri F. Scientific colour maps. Zenodo; 2023. doi:10.5281/zenodo.8409685

63. Crameri F, Shephard GE, Heron PJ. The misuse of colour in science communication. Nat Commun. 2020;11: 5444. doi:10.1038/s41467-020-19160-7

Additional Editor Comments:

The manuscript presents a well-written and comprehensive review of muscle co-contraction indexes. However, please update it by addressing all the concerns raised by the reviewers.

Response: Thank you for your feedback. We have included our response to the reviewer below.

Reviewer’s Comments to Authors:

Reviewer 1

The manuscript is well written and the methodological framework is carefully developed. The comparison of multiple co-contraction indices is particularly interesting and represents a valuable methodological contribution. I only have a minor suggestion regarding Figure 9. While the normalization parameters are mentioned in the manuscript and the script used to generate the synthetic EMG signal is available in the GitHub repository, the intermediate steps that lead from those parameters to the plot shown in Figure 9 are not entirely clear from the text. It would therefore be helpful if the authors could briefly clarify the processing steps that transform the reported normalization parameters into the final signal representation shown in the figure.

Response: Thank you for your review. There are no processing steps as this is synthetic EMG. We simply used 0.6, 1.0, and 1.4 as the maximum height of a linear function (with zero intercept) to create the dotted, dash-dotted, and dashed lines in Figure 9. Bar plots on the left represent the co-contraction index calculated from the solid line vs each dash/dot line. We have updated the caption of figure 9 (excerpt below) to clarify.

Revised Text: Figure 9. Amplitude normalization influences each CCI in distinct ways. In panels A and B, the amplitude of EMG1 (solid black lines) is fixed, while that of EMG2 varies to represent three amplitude normalization conditions: non-normalized (dotted line, maximum amplitude 1.4), within-task normalization (dash-dotted line, maximum amplitude 1), and MVC normalization (dashed line, maximum amplitude 0.6). Panels C and D show bar plots of co-contraction values for all six CCIs, calculated from the three EMG1 and EMG2 combinations in panels A and B. Amplitude-driven CCIs (i.e., Rudolph and Unnithan-Frost) are most affected when amplitude normalization reduces both EMGlow and total EMG (panel C). In contrast, shaped-based CCIs, like Falconer-Winter, are most sensitive when normalization alters only one of EMGlow and total EMG (panel D). Abbreviations: CCISR: simple ratio; CCIFW: Falconer-Winter; CCITS: Thoroughman-Shadmehr; CCIR: Rudolph; CCIUF: Unnithan-Frost; CCIT: temporal.

---

## [Editor Report · Decision Letter 1]

12 May 2026

A comparative analysis of co-contraction indices using synthetic EMG data: Implications for selection and interpretation

PONE-D-26-05502R1

Dear Dr. Sawers,

We’re pleased to inform you that your manuscript has been judged scientifically suitable for publication and will be formally accepted for publication once it meets all outstanding technical requirements.

Kind regards,

Andrea Tigrini, Ph.D.

Academic Editor

PLOS One
---

## [Editor Report · Acceptance letter]

PONE-D-26-05502R1

PLOS One

Dear Dr. Sawers,

I'm pleased to inform you that your manuscript has been deemed suitable for publication in PLOS One. Congratulations! Your manuscript is now being handed over to our production team.

Kind regards,

on behalf of

Dr. Andrea Tigrini

Academic Editor

PLOS One